# Evaluation of Immune Nanoparticles for Rapid and Non-Specific Activation of Antiviral and Antibacterial Immune Responses in Cattle, Swine, and Poultry

**DOI:** 10.3390/ani13101686

**Published:** 2023-05-18

**Authors:** William H. Wheat, Lyndah Chow, Alyssa M. Betlach, Maria Pieters, Jade Kurihara, Cooper Dow, Valerie Johnson, Franklyn B. Garry, Steven Dow

**Affiliations:** 1Department of Clinical Sciences, Colorado State University, Fort Collins, CO 80523, USA; 2Department of Microbiology, Immunology and Pathology, College of Veterinary Medicine and Biomedical Sciences, Colorado State University, Fort Collins, CO 80523, USA; 3Swine Veterinary Center, St. Peter, MN 56082, USA; abetlach@swinevetcenter.com; 4Department of Veterinary Population Medicine, Veterinary Diagnostic Laboratory and Swine Disease Eradication Center, College of Veterinary Medicine, University of Minnesota, St. Paul, MN 55455, USA; 5Idaho State University, Pocatello, ID 83209, USA; cooper.dow@focus.org; 6College of Veterinary Medicine, Michigan State University, East Lansing, MI 48824, USA; john7670@msu.edu

**Keywords:** agriculture, immunotherapy, cytokines, innate immunity, pandemic, countermeasures

## Abstract

**Simple Summary:**

The recent SARS-CoV-2 pandemic has underscored the impact of a single pathogen on public health and economic stability. Similarly, widespread disease outbreaks within agricultural animal populations are potentially devastating by producing food shortages and global starvation. Protecting agricultural animals from viral/bacterial disease outbreaks will depend on the development of countermeasures generating rapid innate immune protection. Accordingly, we have developed a rapidly deployable generalized immune stimulant activating innate immunity at mucosal sites. Past studies show that this platform, termed *LTC* (liposome-TLR complexes), can rapidly activate and recruit innate immune cells in rodents and companion animals. Moreover, prior rodent and cattle studies showed complete protection against lethal viral and bacterial challenge studies. In the studies described herein, we demonstrate that the LTC platform can be extended to demonstrate rapid activation of innate immune modulators from cell lines and immune cells from peripheral blood in three key agricultural animals: cattle, pigs, and poultry. Cell lines and immune cells from peripheral blood from all three of these species demonstrated significant increases in antibacterial and antiviral immune mediators when treated with LTC in vitro. These data provide justification for further testing of this platform in disease challenge models for these animals.

**Abstract:**

Given the rapid potential spread of agricultural pathogens, and the lack of vaccines for many, there is an important unmet need for strategies to induce rapid and non-specific immunity against these viral and bacterial threats. One approach to the problem is to generate non-specific immune responses at mucosal surfaces to rapidly protect from entry and replication of both viral and bacterial pathogens. Using complexes of charged nanoparticle liposomes with both antiviral and antibacterial toll-like receptor (TLR) nucleic acid ligands (termed liposome-TLR complexes or *LTC*), we have previously demonstrated considerable induction of innate immune responses in nasal and oropharyngeal tissues and protection from viral and bacterial pathogens in mixed challenge studies in rodents, cattle, and companion animals. Therefore, in the present study, we used in vitro assays to evaluate the ability of the LTC immune stimulant to activate key innate immune pathways, particularly interferon pathways, in cattle, swine, and poultry. We found that LTC complexes induced strong production of type I interferons (IFNα and IFNβ) in both macrophages and leukocyte cultures from all three species. In addition, the LTC complexes induced the production of additional key protective cytokines (IL-6, IFNγ, and TNFα) in macrophages and leukocytes in cattle and poultry. These findings indicate that the LTC mucosal immunotherapeutic has the capability to activate key innate immune defenses in three major agricultural species and potentially induce broad protective immunity against both viral and bacterial pathogens. Additional animal challenge studies are warranted to evaluate the protective potential of LTC immunotherapy in cattle, swine, and poultry.

## 1. Introduction

Rapidly spreading infectious disease outbreaks within major agricultural animal populations can be devastating to animal health and to economies and lead to food shortages and global starvation. Recent examples include foot and mouth disease and rinderpest in cattle, swine vesicular disease, African swine fever in swine, and avian influenza and Newcastle disease in poultry [1,2,3,4,5]. As examples of the widespread impact of rapidly spreading pathogens, outbreaks of H7N1 highly pathogenic avian influenza (HPAI) in poultry have affected nearly 50 million birds in 21 US states. The 2001 foot-and-mouth and bovine spongiform encephalopathy (BSE) pandemic in the UK cost upwards of USD 10 billion and imposed significant economic hardships on farmers through banned meat exports while causing a significant reduction in tourism [4,6]. As much as one third of the global domestic swine population has been decimated by infection with African Swine Fever Virus (ASFV) [1,7,8]. More recently, HPAI caused by H5 influenza viruses has been increasing significantly in wild birds and poultry globally [9,10].

Typically, at the time of the initial outbreak, the identity of the pathogens may be unknown. More importantly, for several of the pathogens noted above, there are no currently effective vaccines, nor are the vaccines that do exist always widely available globally. In addition, vaccine-induced protection usually takes at least two weeks before effective antibody or T cell responses develop. Thus, there is a window of approximately two weeks or longer during which the target population remains unprotected from pathogen infection. This gap in protection is not fully addressed by conventional countermeasures, such as isolation of affected populations or test and slaughter strategies, which can lead to substantial animal mortality and marked economic losses.

At present, one of the most promising strategies to address this gap in protection may be to rapidly induce non-specific innate immune protection at susceptible, point-of-entry mucosal surfaces, such as the nasal and oropharyngeal surfaces, with their innate immune responsive epithelial, resident and recruited myeloid, and lymphoid cell populations [11]. An ideal non-specific immunotherapeutic possesses several key attributes. These include rapid onset of induction of expansive protection against both viral and bacterial pathogens at key mucosal surfaces, the ability to be delivered as a spray or aerosol for broad dispersal, stability at room temperatures, and potency and low costs to manufacture.

We propose that the potent mucosal innate immunotherapeutic termed liposome-TLR complexes (LTC nanoparticles), developed and previously evaluated for efficacy in companion animals (dogs, cats) and recently in cattle, may serve that role [12,13,14], Liposome-TLR complexes are nanoparticles consisting of cationic liposomes complexed to nucleic acids designed to activate both the TLR3 and TLR9 pathways, for induction of both antiviral and antibacterial immunity. In addition, the LTC particles have been designed to adhere more effectively to epithelial cells by the addition of carboxymethylcellulose. The resulting nanoparticles have a net positive charge to create an additional binding to negatively charged cell surfaces and are of an appropriate size (approximately 250 to 300 nm) to assure uptake by relevant innate immune cells (monocytes, macrophages, dendritic cells) for the induction of innate immune responses. In addition, the use of charged liposomes to introduce the TLR3 and TLR9 nucleic acids intracellularly also assures the activation of other innate immune pathways, such as the cyclic GMP-AMP synthase-stimulator of interferon genes (cGAS-STING) and the retinoic acid-inducible gene-1 (RIG-I) further inducing the production of both type I and II interferons [15,16].

An earlier iteration of these LTC nanoparticles containing only a TLR9 ligand has been widely evaluated in rodent challenge studies with rapidly lethal pathogens. For example, complete or nearly complete protection from pneumonic *Burkholderia mallei* and *B. pseudomallei*, *Yersinia pestis*, and *Francisella tularensis* was achieved by a single intranasal application of LTC nanoparticles [17,18,19,20]. In addition, protection from equine encephalitis virus and Punta Tora virus infection was achieved by parenteral administration of TLC nanoparticles [21].

Thus, there is considerable data from other species to suggest that LTC immunotherapy may be effective against diverse pathogens that reach the lungs via inhalation or mucosal spread. However, these LTC nanoparticles have not been evaluated previously for their ability to induce innate immune responses in other major agriculture species other than cattle, including swine and poultry. Therefore, the present study was designed to assess the induction of key protective cytokines by relevant innate immune cell populations in swine and poultry, using both macrophage cell lines, as well as primary leukocyte cultures. Figure 1 schematically illustrates the method of testing species’ cell lines (Figure 1A) and the methodology used to test innate immune cells ex vivo (Figure 1B) in this study. Cells from cattle were also evaluated in this study as a point of reference, inasmuch as LTC complexes have previously demonstrated protective efficacy from bovine respiratory disease in cattle [22]. The study assessed the induction of interferon and TNFα production and in some cases, direct cell activation. The induction of antibacterial intracellular pathways, including reactive nitrogen and reactive oxygen, was also assessed. Dose-dependent innate immune activation was evaluated in each species, and the relative potency of responses was compared between species to better provide a guide to the relative effectiveness of LTC nanoparticles as a broad cross-species immunotherapy.

## 2. Materials and Methods

### 2.1. Preparation of Liposome-TLR Complexes

Liposomes were prepared as described previously [12,13]. Polyinosinic-polycytidylic acid (poly (I:C)) (InvivoGen, San Diego, CA, USA; cat # tlrl-picw) and synthetic 24-bp double-stranded pCpC-rich DNA oligonucleotide (InvivoGen, San Diego, CA, USA; cat # tlrl-2006) were added to pre-formed liposomes to form liposome-TLR complexes (LTC). The endotoxin content of the plasmid DNA was between 0.04 and 0.25 EU/ug. Carboxy-methylcellulose (Sigma-Aldrich, St. Louis, MO, USA; cat # 419281) was added to the pre-formed complexes to produce the final LTC material for study.

### 2.2. Cell Lines and Antibodies

The bovine nasal turbinate cell line was acquired from the American Type Culture Collection (Manassas, VA, USA) (ATCC # CR-1390). The chicken cell lines HD11 (chicken macrophage cell line) and MQ-NCSU (chicken mononuclear cell line) were a generous gift from Dr. Matthew Koci at North Carolina State University. HD11 cells were cultured at 42 °C in a humidified incubator with 5% atmospheric CO_2_ in Dulbecco’s Modified Eagle Medium (DMEM), 10% fetal bovine serum (FBS), 4 mM L-glutamine, 100 units penicillin, and 100 µg/mL streptomycin. MQ-NCSU cells were cultured at 42 °C as well but in a medium comprised of equal volumes of McCoy’s 5A and Lebovitiz L-15 medium (Gibco/ThermoFisher, Waltham, MA, USA) containing 10% chicken serum, 8% FBS, 1 mM 2-mercaptoethanol and 1% tryptose phosphate broth. The 3D4/21 (swine alveolar macrophage cell line) was purchased from ATCC (CRL-2843) and cultured in complete DMEM containing 10% FBS.

### 2.3. Separation of Peripheral Blood Mononuclear Cells (PBMC)

Bovine, porcine, and avian PBMCs were prepared from whole blood obtained by jugular venipuncture from healthy cattle and swine and collected into EDTA tubes. Blood from chickens was obtained by venipuncture from the brachial wing vein (cutaneous ulnar vein). All studies involving blood collection from healthy animals were approved by the Institutional Animal Care and Use Committee at Colorado State University (Ft. Collins, CO, USA) and the University of Minnesota (St. Paul, MN, USA). For separation of PBMC, blood was diluted 1:2 with sterile PBS, then layered over a Ficoll-Paque™ (Millipore/Sigma, Burlington, MA, USA; cat # GE17-1440-02) gradient and centrifuged. Cells were collected and washed twice in PBS and then resuspended in a complete tissue culture medium. Cells were plated in either 24- or 96-well flat bottom plates (CellTreat, Pepperell, MA, USA) at a density of 4 × 10^6^ cells/mL in either 200 μL of medium for 96-well plates or 1 mL for 24-well plates. For assays involving PBMC activation, LTCs were added at 3–4 different dilutions (1.0 µL/mL, 10 μL/mL, 25 μL per well, and 50 μL per well) in triplicate wells of PBMC in 100–1000 μL complete medium, with careful mixing, and the cells were then incubated at 37 °C for a total of 48 h.

### 2.4. Generation of Primary Cultures of Bovine Monocyte-Derived Macrophages (MDM)

Blood was obtained from healthy cattle and plated at 1 mL in triplicate wells per animal at a concentration of 3.0 × 10^6^ per well. Monocytes from the PBMC were allowed to adhere to the plastic wells for 4 h at 37 °C, and non-adherent leucocytes were removed by gently decanting and washing the wells with warm PBS. Complete DMEM + 15%FBS containing 30 ng/mL of cross-reactive recombinant human macrophage colony-stimulating factor (rhM-CSF) (PeproTech, Rocky Hill, CT, USA, NJ cat # 300-25) was added to the adherent monocytes. Every 3 days, 50% of the medium was removed and replaced with fresh rhM-CSF-containing medium. After a total of 7 days of differentiation/proliferation, adherent macrophages were either untreated or treated with either LPS or LTC at the indicated concentrations for 36h, and supernatants were analyzed for species-specific cytokine/chemokine release by ELISA or release of NO (see below) or ROS (see below). For analysis of cell surface major histocompatibility complex II (MHCII) expression, treated cells were induced to detach from culture wells by treatment with ice-cold PBS containing 5 mM EDTA for 20 min. Released cells were washed and stained with a FITC-conjugated anti-bovine MHCII (Bio-Rad, Hercules, CA, USA) analyzed by flow cytometry.

### 2.5. Assessment of NO Production

Mononuclear cells from PBMC or MDM from cattle were seeded in triplicate wells of 24-well plates at a cell density of 2–4 × 10^6^ cells per well in 1 mL and activated with LTC for 36 or 48 h (positive controls were wells treated with LPS). Nitric oxide release was determined by analysis of culture supernatants by the Griess reagent assay using a prepared kit (Promega, Madison, WI, USA; cat # G2930). Nitric oxide release was assessed by measuring the formation of a NO breakdown product, nitrate (NO_2_^−^), using a colorimetric assay.

### 2.6. Assessment of ROS Generation

PBMC or bovine MDM were cultured and treated with either LPS or LTC as described above, prior to adding a freshly prepared 100 µM solution of the cell-permeant 2′, 7′-dichlorodihydrofluorescein (H_2_DCFDA; Milllipore/Sigma, Burlington, VT; cat# D6883) in complete DMEM to the cells. H_2_DCFDA was allowed to load into the cells for 15 min at 37 °C. Cells were subsequently activated for an additional 30 min with either LPS or LTC to stimulate ROS production. Cells were then washed and resuspended in fluorescence-activated cell sorting (FACS) buffer (PBS + 2% FBS + 0.09% sodium azide) and analyzed by flow cytometry for fluorescein gMFI generated by the oxidized H_2_DCFDA by ROS.

### 2.7. ELISA Assay for Measurement of Cytokine Production

Cell culture supernatants were analyzed by ELISA using the following species-specific kits. ELISA kits: Bovine multiplex Luminex^®^ kits for multiplex analysis of INFγ, IL-17A, IL-8, MCP-1, IL-10, IL-1α, IL-36RA, IL-6, and TNFα were from Millipore-Sigma (St. Louis, MO, USA). Chicken IL-6 from ThermoFisher (Cat # ECH6RB), Chicken INFγ from ThermoFisher (Cat # ECH2RB), and Chicken IFNα from ThermoFisher (Cat # ECH1RB). Porcine ELISA kits for IFNα (cat # ES7RB) and IFNβ (cat # ES8RB) were purchased from Invitrogen (Waltham, MA, USA). All ELISA assays were performed as per the manufacturer’s recommendations.

### 2.8. RT-PCR Assays for Measurement of IFNβ Production by Cells from Poultry

Briefly, cDNA was prepared from either cell lines or PBMC following isolation/purification of mRNA from recovered cells, which was subsequently reverse transcribed using a commercial kit (Qiagen, Germantown, MD, USA). Cellular cDNA was amplified using a qPCR MX300p system instrument (Agilent, Santa Clara, CA, USA). Primer sequences for bovine IFNβ were obtained from BLAST searches, and qRT-PCR was used to quantify transcript levels, as reported previously [14]. The chicken (*Gallus gallus*) interferon beta mRNA sequence was obtained from GenBank (accession/version # AY974089.1). Primers for intercalating qRT-PCR were obtained by entering the mRNA sequence into the Integrated DNA Technologies PrimerQuest tool https://www.idtdna.com/pages/tools/primerquest (accessed on 13 June 2021) (Forward-CTTCGTAA-ACCAAGGCACGC) and (Reverse-ATGGTCCCAGGTACAAGCAC), resulting in an amplicon length of 130 bp.

### 2.9. Statistical Analysis

Graphical analysis was performed using GraphPad PRISM Software, La Jolla, CA, USA. Significant differences between cytokine/chemokine or NO/ROS output were determined using a one-way ANOVA with *p*-values denoted in each figure legend.

## 3. Results

### 3.1. Immune Response of Bovine Nasal Turbinate Epithelial Cells to Treatment with LTC

Previously, we showed that the LTC nanoparticles activated bovine PBMC and induced local immune responses and protective immunity to a mixed pathogen challenge model of bovine respiratory disease (BRD) in cattle following intranasal administration [22]. In the present study, we expanded those investigations to also include the induction of innate immune responses by turbinate epithelial cells, an early target cell for viral entry into the respiratory tract [23,24,25,26]. Treatment of bovine nasal turbinate cells with LTC nanoparticles for 36 h stimulated a significant, concentration-dependent increase in the secretion of the antiviral and antibacterial cytokines IL-6 (Figure 2A) and TNFα (Figure 2B). The paradoxical decrease in TNFα production at the highest LTC concentration evaluated (50 µL/mL) may have resulted from the known cytotoxicity of liposome-TLR complexes in vitro [22].

### 3.2. In Vitro Treatment of Bovine PBMC with LTC Stimulated Significant Increases in Protective Innate Immunity Mediators

Treatment of bovine PBMC with LTC significantly stimulated increased secretion of key antiviral and antibacterial cytokines, including IFNγ, TNFα, IL-6, IL-17A, and IL-1α ([22] and Figure 3A–E). Additionally, LTC stimulated significant increases in the production of IFNα (Figure 3F). Expression of IFNβ1 mRNA was also increased as assessed by qRT-PCR (data not shown). Collectively, the ability to stimulate the rapid production of a broad array of pro-inflammatory cytokines and Type I interferons with both antiviral and antibacterial activity suggests that LTC nanoparticles can induce broad protective immunity in cattle.

The induction of LTC production of other cytokines in cattle was assessed using a bovine multiplex ELISA assay. This assay found variable induction states for IL-36RA, MCP-1, and IL-8 by LTC (Figure 3G–I, respectively), along with decreased production of the anti-inflammatory cytokine IL-10 (Figure 3J).

### 3.3. Treatment of Bovine Monocyte-Derived Macrophages (MDM) with LTC Stimulated Multiple Innate Immune Pathways

The ability of LTC to activate macrophages in cattle was assessed using in vitro-generated macrophages from adherent monocytes (monocyte-derived macrophages, MDM). The MDM cultures were treated with either 20 ng bovine IFNγ or LPS (300 ng/mL) as positive controls or 10, 25, and 50 µL LTC/mL for 48 h and supernatants were harvested for cytokine, NO, and ROS analysis. The adherent cells were then detached and evaluated for activation by measuring the upregulation of MHCII expression by flow cytometry. Flow cytometric analysis of MDM demonstrated a dose–response increase in MHCII expression following treatment with LTC nanoparticles (Figure 4A,B). Cytokine analysis demonstrated the significant induction of the secretion of TNFα and IL-6 (Figure 4C,D).

Additionally, LTC-treated MDM were assessed for the production of NO and ROS. Most concentrations of LTC evaluated triggered significant increases in the production of NO and ROS (Figure 4E,F). Collectively, these findings indicate that LTC can directly activate bovine macrophages for antiviral and antibacterial activity.

### 3.4. Porcine Macrophage Responses to LTC Stimulation

Next, we used a porcine macrophage cell line to assess the ability of swine to respond to TLC complexes. These studies utilized the porcine alveolar macrophage cell line as a readout, and the cells were treated with increasing concentrations of LTC (10, 25, and 50 mL/mL) for 24 h, as described in Methods. Treatment of porcine macrophages with increasing concentrations of LTC induced significant increases in the secretion of IFNα and IFNβ (Figure 5A,B). This response included significant increases in IFNα release at the lowest LTC concentration evaluated (10 µL/mL) with greater increases in IFNα production with higher concentrations (25 and 50 µL/mL) (Figure 5A). Similarly, treatment of porcine macrophages with LTC also significantly stimulated increased secretion of IFNβ (Figure 5B). Notably, treatment of these macrophages with the TLR4 agonist LPS failed to elicit any significant increase in IFNα production (Figure 5A,B). When we compared the relative responsiveness of porcine versus bovine macrophages to LTC-mediated INFα elevation, we found that, in general, porcine macrophages were roughly equivalent in their responses to LTC stimulation as bovine macrophages (e.g., compare Figure 3F to Figure 5A), although these results must be interpreted cautiously due to differences in the macrophage populations being studied (i.e., primary monocyte-derived vs. alveolar macrophage cell line).

We next assessed the response of porcine macrophages to LTC stimulation in triggering the production and release of NO (Figure 5C). We found that all three concentrations of LTC evaluated triggered significant NO responses, with a concentration-dependent increase. Additionally, significant upregulation of intracellular ROS responses was also observed, and the response was also LTC concentration-dependent (Figure 5D).

### 3.5. Responses of Porcine Leukocytes to LTC Activation

Porcine PBMC isolated from healthy swine were treated with LTC to assess their cytokine responses and to compare these responses to those elicited in cattle PBMC (compare Figure 3 and Figure 6). We found that LTC treatment triggered significantly increased secretion of IFNα at the three highest doses evaluated (Figure 6A). IFNβ was also significantly upregulated in LTC-treated PBMC (Figure 6B). Overall, the level of LTC-mediated IFNα secretion was greater than that observed for IFNβ. Interestingly, treatment of porcine PBMC with LPS also failed to elicit any significant increase in IFNα production, similar to the lack of response to LPS by the porcine macrophage cell line (Figure 5A and Figure 6A).

Assessment of PBMC release of NO and ROS treated with LTC showed significant increases in both innate immune mediators in a concentration-dependent fashion for NO (Figure 6C) or ROS (Figure 6D). These mediators were similarly released and increased as a dose-response, as was observed in bovine macrophages in this work above as well as for bovine PBMC in a previous study (Figure 4E,F [22]). These studies also indicate significant increases in the release of NO and ROS from either myeloid-derived macrophages or cell lines when compared to release from PBMC.

### 3.6. LTC Responses by Macrophages from Poultry

We next assessed the capacity of chickens to respond to LTC activation. These initial studies evaluated two different macrophage cell lines from chickens (HD11 cells and MQ-NCSU cells). The two cell lines were characterized previously and differ in that HD11 is a macrophage line derived from chicken bone marrow and transformed by the avian leukemia virus, whereas MQ-NCSU is a macrophage line derived from spleen and transformed with a strain of Marek’s disease virus [27,28]. We observed that HD11 macrophages responded to LTC activation with a significant release of IFNα (Figure 7A). In addition, LTC treatment upregulated the expression of INFβ mRNA as assessed by qRT-PCR (Figure 7B). Treatment of HD11 macrophages with increasing doses of LTC in vitro also stimulated significant IL-6 cytokine release (Figure 7C). Production of NO and ROS was also significantly increased when HD11 cells were treated with LTC when compared to untreated cells. Increased NO production appeared as a dose-response to LTC concentration, whereas ROS production was stimulated similarly in all treatment concentrations (Figure 7D,E).

Innate immune responses to LTC were also assessed in the chicken MQ-NCSU macrophage cell line [28] (Figure 8). As observed with HD11 cells, LTC treatment of MQ-NCSU cells increased INFα secretion yet decreased but remained elevated at higher LTC concentrations when compared to untreated cells (Figure 8A). Treatment of MQ-NCSU cells also resulted in an approximately two-fold increase in INFβ in mRNA expression (Figure 8B). LTC Treatment of MQ-NCSU cells resulted in a greater relative increase in IFNβ transcription than treatment with LPS (Figure 8B). As observed for the HD11 cells above, treatment of MQ-NCSU cells with increasing doses of LTC stimulated significant increases in IL-6 cytokine release in vitro (Figure 8C). Interestingly, treatment of MQ-NCSU with LTC resulted in a robust dose-responsive increase in NO production but failed to stimulate increased ROS production, a marked difference compared to the HD11 line (Figure 8D,E). Thus, overall, two different macrophage cell lines from chickens responded strongly to LTC nanoparticle activation, with small differences in individual responses. When compared to macrophages from cattle and from swine, we noted that macrophages from poultry overall (with the notable exception of the ROS response by the MQ-NCSU) appeared to be similarly responsive to LTC activation. The significant difference between ROS produced in LTC-treated HD11 [27] and MQ-NCSU [28] cell lines is likely due to the fact that the HD11 line is more macrophage-like, whereas the MQ-NCSU line was more monocytic in phenotype and may not fully differentiate to an oxygen radical generating cell.

### 3.7. Response of PBMC from Chickens to LTC Activation

Treatment of PBMC prepared from the blood of healthy chickens with LTC stimulated significant increases in the release of INFγ when compared to untreated cells (Figure 9A). Additionally, higher concentrations of LTC (10 and 25 µL/mL) stimulated a significant release of antibacterial and antiviral IFNα. Release of IFNα was generally greater with LTC activation than when PBMC were treated with 300 ng/mL LPS (Figure 9B). Likewise, secretion of IL-6 was triggered by LTC treatment as well (Figure 9C).

Treatment of chicken PBMC also significantly upregulated the release of NO (Figure 9D). Stimulation of PBMC with LPS also triggered high levels of NO release, indicating that the LPS used in these studies was active (Figure 9D). Likewise, treatment of chicken PBMC with higher concentrations of LTC stimulated significant ROS production, indicative of a robust antibacterial response. Stimulation of PBMC with LPS did not produce any significant production of ROS when compared to untreated controls (Figure 9E). Overall, we found comparatively similar dose responses and overall percentage increases of IFNα release from PBMC from all three species; the general release of this Type I IFN was to a much lower extent in the swine (compare Figure 3F, Figure 5A and Figure 9B). Increased percentages of NO release between swine and poultry were similar, with a noted significantly higher overall increase in NO on a per cell basis for swine PBMC (compare Figure 6C with Figure 8D) Both swine and chicken PBMC released ROS similarly and as a dose-response, with the chicken PBMC releasing more ROS on a per-cell basis (compare Figure 6D with Figure 9E).

## 4. Discussion

There currently exists a considerable unmet need for new immunotherapeutic approaches to rapidly and broadly protect major agriculture species (cattle, swine, and poultry) from threats posed by highly infectious and pathogenic viruses and bacteria. In the studies reported here, our key findings were that LTC nanoparticles were potent activators of multiple different innate immune pathways in all three species evaluated. The LTC complexes were particularly effective in inducing the production of IFNα and IFNβ, which is important because type I IFNs are key early mediators of both antiviral and antibacterial immunity. We also found that immune cells from porcine and poultry were comparatively more or less responsive to LTC activation when compared to bovine, depending on which factors were examined, although all three species responded robustly. However, it was somewhat surprising that two key proinflammatory chemokines (IL-8 and MCP-1) were paradoxically not induced in vitro in the multiplex analyses of bovine PBMC. It is, however, noteworthy that of all 10 of the mediators measured in the multiplex analyses, these two were chemokines whose function is to induce cell migration as opposed to cytokines which are in situ mediators of pro- or anti-inflammation. It is likely that optimal induction of chemokines requires full immune context in vivo that is not represented in the small confine and simplicity of the in vitro environment. Accordingly, we reported increases in mRNA encoding both IL-8 and MCP-1 in cells purified from nasopharyngeal swabs at times for post-intranasal LTC treatment in cattle [22].

Given the imminent threat potentially imposed by numerous emerging or re-emerging diseases, it has become important to develop methods to rapidly generate non-specific immunity to diverse pathogens. Effectively activating immune defenses at mucosal respiratory and oral routes of pathogen entry by engagement of host innate immune defenses is currently the most plausible strategy for generating that type of protection. Such measures would provide crucial time for the development of vaccines or other more pathogen-specific therapeutics that would take considerably more time to develop. In some cases, non-specific innate immune activation alone may be sufficient to protect from disease or lessen disease severity [11,29,30]. In the recent study, we demonstrated in proof-of-concept studies in cattle that intranasal application of a non-specific immunotherapeutic provided significant protection from clinical illness and mortality from bovine respiratory disease due to mixed pathogen challenge [22].

In the current study, we found that the immune cells of cattle, swine, and poultry all responded immunologically to activation with LTC nanoparticles. For example, both macrophages and PBMC from cattle, pigs, and chickens all produced key antiviral and antibacterial cytokines and immune mediators, NO and ROS, upon LTC activation. Indeed, in all three species, LTC significantly increased the secretion of critical antiviral innate immune mediators, including type I IFNs. Production of type I IFNs is important because, in many viral and bacterial infections, these are the key gateway cytokines regulating immune defenses in epithelial and immune target cells [29,31,32].

Finally, the production of NO and ROS by LTC stimulation was significantly increased in PBMC and/or macrophages from all three species. Innate immune cells, including neutrophils and macrophages, utilize NO and ROS as a very efficient first line of defense against harmful pathogens [33]. Production of NO is critical for the initiation of inflammatory responses and activation of innate and adaptive immunity [34,35,36,37]. In addition, ROS molecules play an important role in innate immunity and generate broad pathogen killing in both intracellular and extracellular locations [38,39,40].

## 5. Conclusions

In summary, this study demonstrates the potential for an LTC-based immunotherapy platform to be used as a broadly active, non-specific defense in the three primary agriculture species in the US and globally. This work conclusively demonstrates that immune cell lines and PBMC from cattle, swine, and poultry can generate significant innate immune responses to treatment with LTC. These studies have generated crucial proof-of-concept data in the evolving process of development of the LTC platform as a front-line countermeasure deployed to prevent the spread of disease in the agricultural setting. Pathogen challenge and field studies will be required to fully demonstrate the utility of the LTC platform, although promising recent results in a cattle BRD model suggest a reason for optimism. Additional challenges will include efficient deployment and delivery of such an immune stimulant and demonstration of activity under realistic field conditions. If shown effective in these studies, the LTC platform has the potential to be widely utilized as part of a multi-pronged approach to disease development, management, prevention, and/or reduction by the global agricultural community.

## Figures and Tables

**Figure 1 animals-13-01686-f001:**
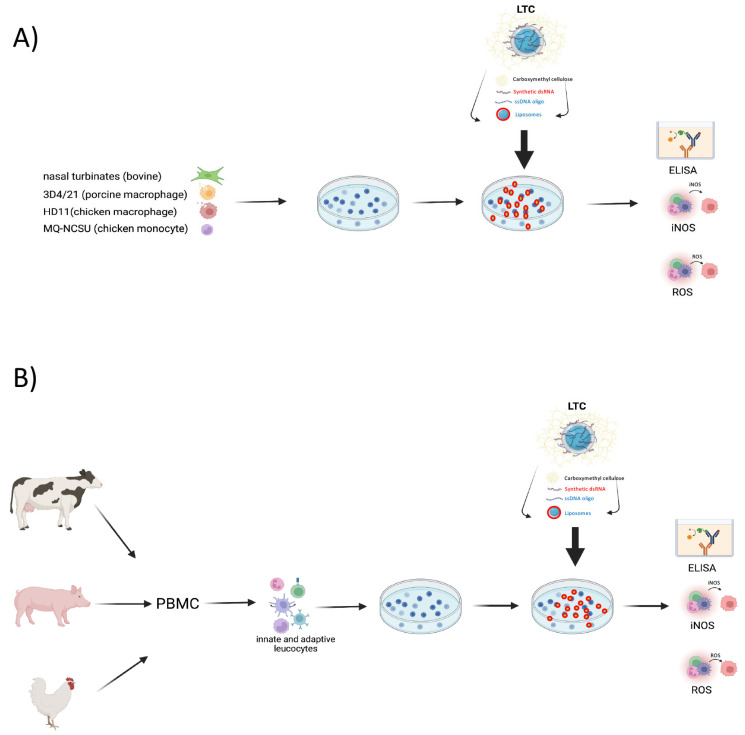
In vitro evaluation approach for evaluation of LTC immune activity in three target animal species. Representative cell lines (**A**) or purified leucocytes (**B**) from whole blood were obtained from dairy or beef cattle, pigs, or chickens. Cell lines or leucocytes were plated among 4.0–6.0 × 10^6^ cells per well in 24 well plates (1 mL per well) and treated for 24 or 36 h with either 1, 5, 10, 25, or 50 µL/mL with LTC nanoparticles as described in Methods. Cells were subsequently harvested and assessed for release of chemokines/cytokines by enzyme-linked immunosorbent assay (ELISA) or culture supernatants were analyzed for inducible nitrous oxide synthase (iNOS) activity and subsequent nitric oxide (NO) production via the Griess Reagent assay, or activated cells were treated with oxidized fluorescein (H_2_DCFDA) and assessed for production of reactive oxygen species (ROS) by flow cytometry.

**Figure 2 animals-13-01686-f002:**
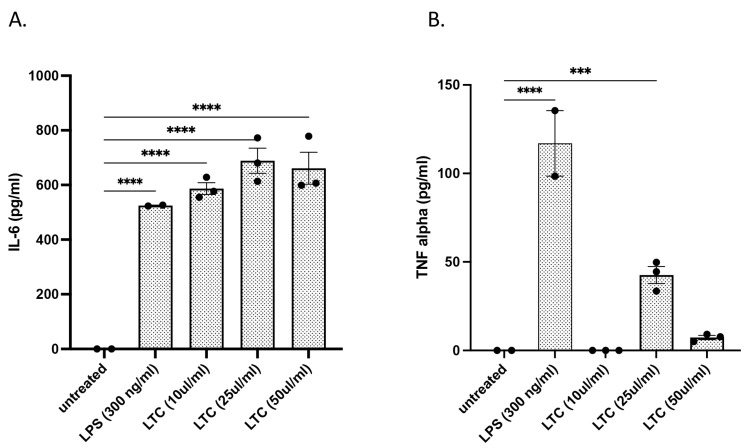
Treatment of bovine nasal turbinate cells (ATCC # CRL-1390) with increasing dosages of LTC stimulates significantly higher secretion of pro-inflammatory cytokines IL-6 and TNFα. Turbinate cells were seeded in 24-well plates and treated with the indicated concentrations of LPS or LTC for 36 h. Secretion of IL-6 (**A**) and TNFα (**B**) was determined by ELISA. Significant differences were determined using a one-way ANOVA with *p*-values of **** ≤ 0.001 and *** ≤ 0.005.

**Figure 3 animals-13-01686-f003:**
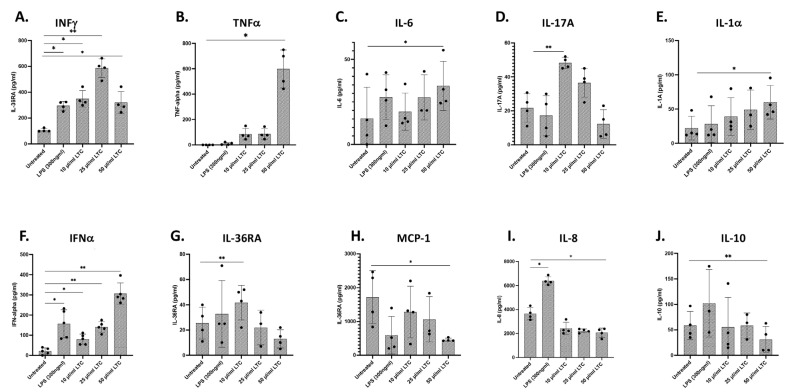
Treatment of bovine PBMC with LTC stimulates the production of pro-inflammatory cytokines with potential potent antiviral and antibacterial activity. Bovine PBMC were plated at 4.0 × 10^6^ cells per well in 24-well tissue culture plates and treated for 36 h. Supernatants were subsequently harvested and assayed for cytokine/chemokine release. ELISA for release of IFNγ (**A**), TNFα (**B**), IL-6 (**C**), IL-17A (**D**), IL-1α (**E**), IFNα (**F**), IL-36RA (**G**) MCP-1 (**H**), IL-8 (**I**), and IL-10 (**J**). Significant differences in cytokine/chemokine release were determined using one-way ANOVA with *p*-values of ** ≤ 0.01 and * ≤ 0.05.

**Figure 4 animals-13-01686-f004:**
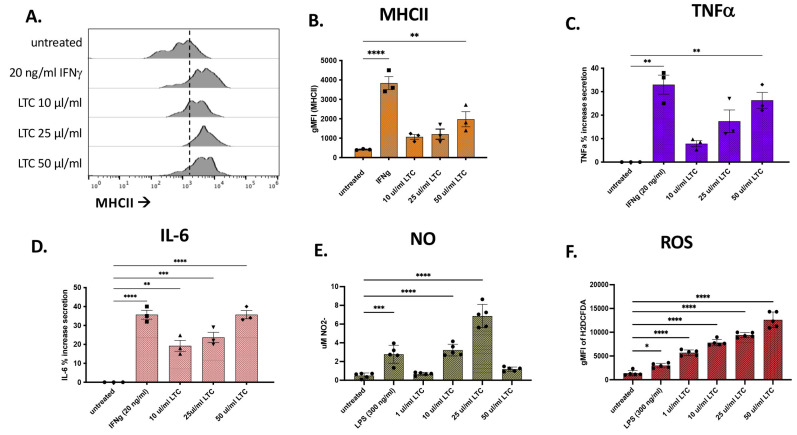
Treatment of bovine primary macrophages with LTC stimulates upregulation of MHCII and release of TNFα, IL-6, NO, and ROS. Bovine MDM were either untreated or treated with either INFγ, LPS, or 10, 25, and 50 ul/mL of LTC for 36 h followed by staining with a FITC-conjugated antibody against bovine MHCII and analyzed for differential cell surface expression by flow cytometry. Raw flow cytometry histograms from flow cytometric analysis of MDM shows differential FITC gMFI (**A**). Bar charts showing a summary of expression of MHCII in MDM from 3 separate animals (**B**). Cell supernatants were assayed by ELISA for TNFα (**C**) and IL-6 (**D**). Positive controls for assessment of NO and ROS in MDM were cultures treated with 300 ng/mL LPS. Supernatants were assayed by the Griess reagent assay to determine the release of NO (**E**). MDM were subsequently treated with 50 µM H_2_DCFDA and analyzed by flow cytometry to determine the extent of ROS production (**F**). Significant differences in MHCII expression, cytokine, and NO release and production of ROS were determined using a one-way ANOVA with *p*-values of **** ≤ 0.001, *** ≤ 0.005, ** ≤ 0.01, and * ≤ 0.05.

**Figure 5 animals-13-01686-f005:**
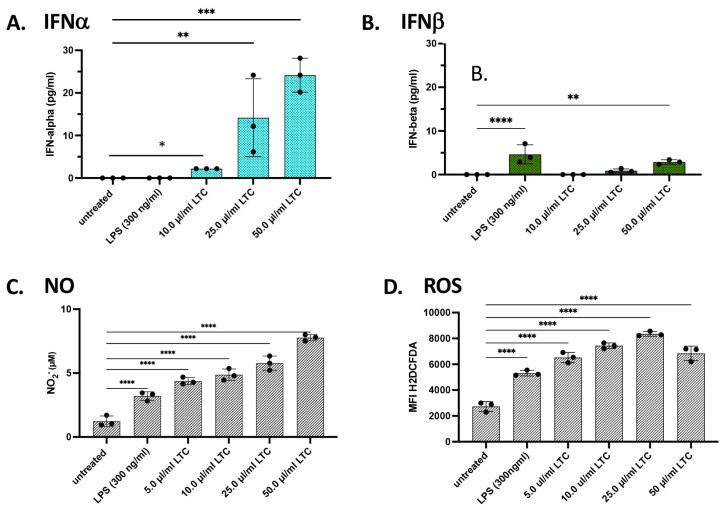
Treatment of the porcine alveolar macrophage cell line 3D4/21 with LTC stimulates the release of the Type I interferons, IFNα, IFNβ, NO, and ROS. Porcine macrophage cells were treated with 10, 25, or 50 µL/mL LTC or with LPS (300 ng/mL) for 36 h. Supernatants were assayed by ELISA for the release of IFNα (**A**) and IFNβ (**B**). Cells were plated and treated as in Figure 3 above, and supernatants were collected and analyzed for NO release by the Griess reagent assay (**C**). Cells were treated with 10 µM H_2_DCFDA to determine the release and activation of ROS (**D**). Significant differences were determined using a one-way ANOVA with *p*-values of **** ≤ 0.001, *** ≤ 0.005, ** ≤ 0.01, and * ≤ 0.05.

**Figure 6 animals-13-01686-f006:**
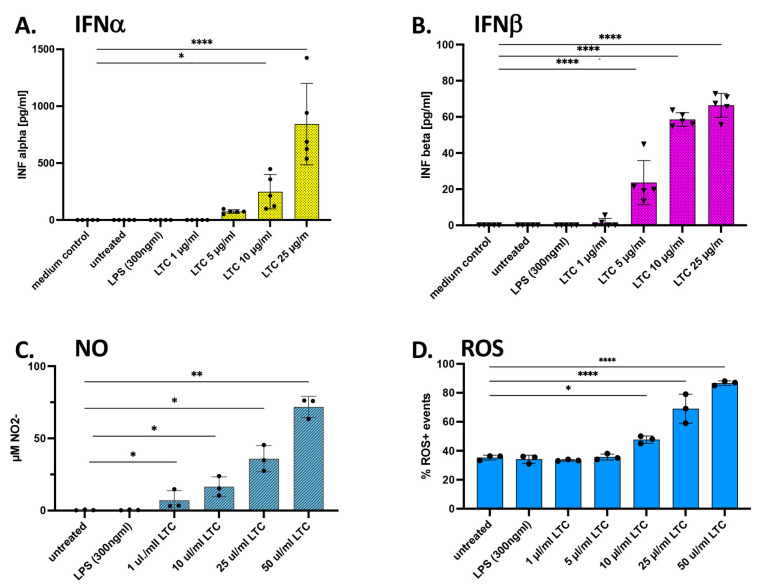
LTC treatment of porcine PBMC stimulates the significant release of IFNα, IFNβ, NO, and ROS. Cultures of PBMC were seeded at 4.5 × 10^6^ cells per well in 24-well plates and treated for 36 h with either nothing (untreated) or 1, 5, 10, or 25 µg/mL of LTC, or 300 ng/mL LPS. Supernatants were assayed by ELISA for the release of porcine IFNα (**A**) or IFNβ (**B**). Culture supernatants were assayed for NO release by the Griess reagent assay (**C**) and for the production of ROS by treatment with 10 µM H_2_DCFDA (**D**). Medium controls were included to rule out any detection of cytokines in the growth medium. Significant differences in Type I IFN, NO, and ROS were determined using a one-way ANOVA with *p*-values of **** ≤ 0.001, ** ≤ 0.01, and * ≤ 0.05.

**Figure 7 animals-13-01686-f007:**
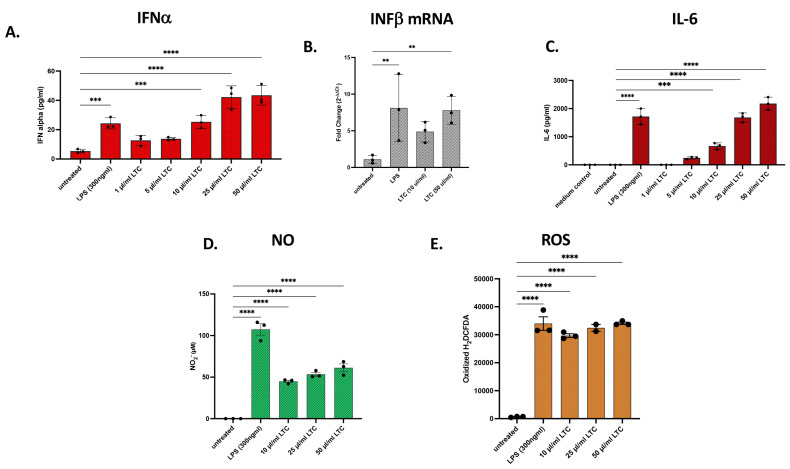
Treatment of chicken macrophage cell line HD11 with LTC stimulated significant increases in the release of IFNα, IL-6, NO, and ROS. Chicken HD11 macrophages were plated in 24-well plates and treated with either nothing, LPS (300 ng/mL), or 1, 5, 10, 25, or 50 µL/mL LTC for 36 h. LTC treatment stimulated significant increases in chicken IFNα release and INFβ mRNA expression from HD11 cells (**A**,**B**). Primer pairs for reverse transcribing and generating cDNA from chicken IFNβ mRNA transcripts were amplified by qRT-PCR. Quantitative comparisons of transcripts encoding IFNβ were determined and compared in either untreated, LPS-treated, or LTC-treated HD11 (**B**). Differences in IL-6 production were determined by chicken IL-6 ELISA (**C**). NO release (**D**) and ROS production (**E**) were assessed using the Griess reagent and H_2_DCFDA assays as described in Methods. Significant differences in the release of these mediators were determined using a one-way ANOVA with *p*-values of **** ≤ 0.001, *** ≤ 0.005, and ** ≤ 0.01.

**Figure 8 animals-13-01686-f008:**
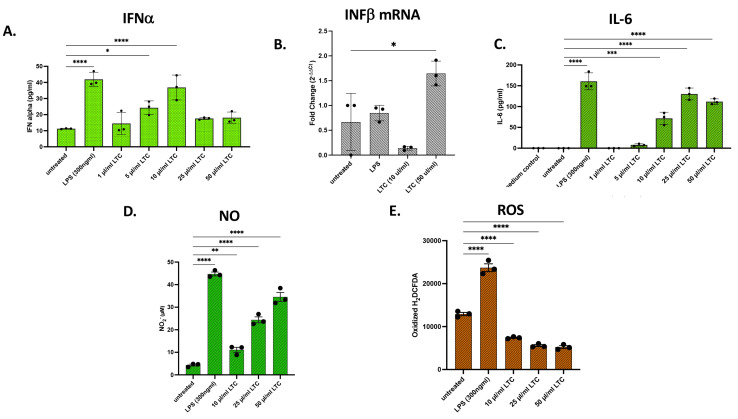
Treatment of the chicken macrophage cell line MQ-NCSU with LTC stimulated significant increases in the release of IFNα, IL-6, and NO but inhibited ROS production. The chicken early monocyte/macrophage cell line MQ-NCSU cells were plated in 24-well plates and treated with 1, 5, 10, 25, or 50 µL/mL LTC or LPS (300 ng/mL) for 36 h, and supernatants assayed for cytokine and NO/ROS production. LTC treatment stimulated significant increases in IFNα release and INFβ mRNA expression (**A**,**B**). IFNβ transcription was assessed in either untreated, LPS-treated, or LTC-treated MQ-NCSU cells (**B**). IL-6 production was determined by specific IL-6 ELISA (**C**). NO release (**D**) and ROS production (**E**) were assessed using the Griess reagent and H_2_DCFDA assays as described above. Significant differences in the release of cytokines, mRNA expression, NO, or ROS were determined using a one-way ANOVA with *p*-values of **** ≤ 0.001, *** ≤ 0.005, ** ≤ 0.01, and * ≤ 0.05.

**Figure 9 animals-13-01686-f009:**
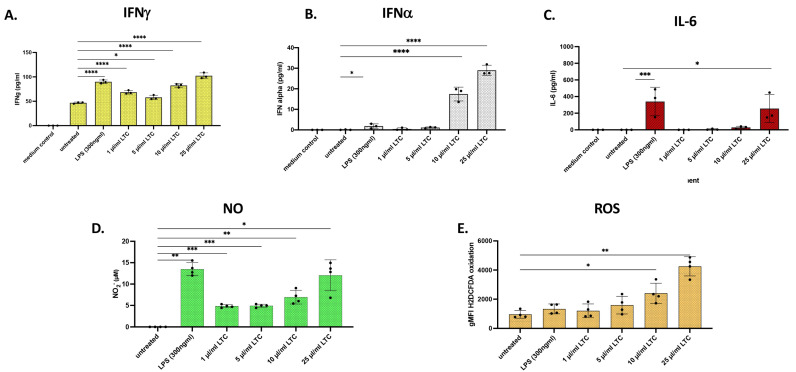
Chicken PBMC treated with LTC demonstrated significant increases in the secretion of IFNγ, INFα, IL-6, NO, and ROS. PBMC from chickens were seeded in 24-well plates in 1 mL cultures at 4.0 × 10^6^ cells per well. Cells were either untreated or treated with either 300 ng/mL LPS, 1, 5, 10, and 25 µL/mL LTC for 36 h at 42 °C. Medium control assays for cytokines were performed in cultures without cells to determine any cross-reaction of the ELISA components to the medium used in the cultures. Specific chicken cytokine ELISA was performed for IFNγ (**A**), IFNα (**B**), and IL-6 (**C**). NO (**D**) and ROS (**E**) were assessed using the Griess reagent and H_2_DCFDA assays as described above. Symbols (●) for each bar represent the mean of triplicate cultures from each of 3–4 different animals. Significant differences in cytokine production via ELISA, NO, or ROS were determined using a one-way ANOVA with *p*-values of **** ≤ 0.001, *** ≤ 0.005, ** ≤ 0.01, and * ≤ 0.05.

## Data Availability

Not applicable.

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
