# Peer review of "Evaluation of Immune Nanoparticles for Rapid and Non-Specific Activation of Antiviral and Antibacterial Immune Responses in Cattle, Swine, and Poultry"

_animals, 2023, doi:10.3390/ani13101686_

Round 1

Reviewer 1 Report

The authors reported the effects of liposome-TLR complexes (LTC) on cell lines originated from macrophages or purified leukocytes of farm animals (beef cattle, pigs and chickens) in terms of innate immune responses, by measurement of cytokines, such as type I- and II- INFs, TNFa, IL-6 and so on, NO release and ROS production,  and demonstrated that TLC, in general, stimulated the innate immune responses of these animals. The study has significance in the field, the methods used were straightforward and experimental designs were sound. However, some explanations in the M&M section were not sufficient, and also some additional explanations and interpretations are necessary in the discussion section in about some of the results.

Specific points

- Characters in Figure 1 are too small to read. Please increase the sizes of the characters in the figure. Also, pictures of LTC are too small to read the ingredients of LTC. Is this on purpose in order to obscure the detailed information of LTC? If so, please disclose the need for proprietary information, and remove such information from the figure. If not, please increase the size of the LTC picture, so that the ingredients can be seen.

-There is no description about the LTC production in the M&M section. Please describe how the LTC used in this study was produced.

-There is no description of statistical analysis in the M&M section. Please explain briefly the methods and the tools used for the data in this paper.

-In Figure 3 H and I, MCP-1 and IL-8 were not induced with LTC treatments, but there is no explanation about the results. The authors should explain the results regarding whether the results are reasonable or not in the discussion section, possibly with helpful references.

-Likewise above, in Figure 8 E, ROS generation was not induced with LTC treatments but no significant discussion about the result. Please explain any possible reason for the results with supporting references. 

Author Response

Reviewer # 1:

The authors reported the effects of liposome-TLR complexes (LTC) on cell lines originated from macrophages or purified leukocytes of farm animals (beef cattle, pigs and chickens) in terms of innate immune responses, by measurement of cytokines, such as type I- and II- INFs, TNFa, IL-6 and so on, NO release and ROS production,  and demonstrated that TLC, in general, stimulated the innate immune responses of these animals. The study has significance in the field, the methods used were straightforward and experimental designs were sound. However, some explanations in the M&M section were not sufficient, and also some additional explanations and interpretations are necessary in the discussion section in about some of the results.

Specific points

- Characters in Figure 1 are too small to read. Please increase the sizes of the characters in the figure. Also, pictures of LTC are too small to read the ingredients of LTC. Is this on purpose in order to obscure the detailed information of LTC? If so, please disclose the need for proprietary information, and remove such information from the figure. If not, please increase the size of the LTC picture, so that the ingredients can be seen.

Figure 1 has been revised with larger characters which should make it easier to read.

-There is no description about the LTC production in the M&M section. Please describe how the LTC used in this study was produced.

We have included a new M&M section (2.1) briefly describing our LTC preparation and citing more detailed methodology therein.

-There is no description of statistical analysis in the M&M section. Please explain briefly the methods and the tools used for the data in this paper.

We have included a new M&M section (2.9) providing a brief description of the software used to determine significant differences between cellular output. Additionally, a description and P results are stated in each Figure legend.

-In Figure 3 H and I, MCP-1 and IL-8 were not induced with LTC treatments, but there is no explanation about the results. The authors should explain the results regarding whether the results are reasonable or not in the discussion section, possibly with helpful references.

We have included a small section in the Discussion section that addresses this discrepancy. (Lines 407-411)

j-Likewise above, in Figure 8 E, ROS generation was not induced with LTC treatments but no significant discussion about the result. Please explain any possible reason for the results with supporting references.

We added a couple of sentences including references in the Results section (lines 401-405) discussing a probable reason for the description between the 2 cell lines.

Reviewer 2 Report

Evaluation of Immune Nanoparticles for Rapid and Non-Specific Activation of Antiviral and Antibacterial Immune Responses in Cattle, Swine and Poultry" by Wheat,W.H. et al describes the use of immune stimulating nanoparticles for developing broad spectrum mucosal immune protection against bacterial and viral diseases.

Please address the following comments

1.Please include catalog nos. for all reagents and tissue culture products used in the Materials and Methods section.

2. Please describe in detail the preparation of LTC and add information on the ligand concentration in the LTC used.

3. Have the authors checked for any difference in the in vitro bacericidal or virucidal property of PBMC/monocytes after treatment with these LTC?

Author Response

Reviewer # 2:

Comments and Suggestions for Authors

Evaluation of Immune Nanoparticles for Rapid and Non-Specific Activation of Antiviral and Antibacterial Immune Responses in Cattle, Swine and Poultry" by Wheat,W.H. et al describes the use of immune stimulating nanoparticles for developing broad spectrum mucosal immune protection against bacterial and viral diseases.

Please address the following comments

      1.   Please include catalog nos. for all reagents and tissue culture products   used in the Materials and Methods section.

We’ve included catalog numbers for specific non-redundant or specifically used items in the M&M.

  1. Please describe in detail the preparation of LTC and add information on the ligand concentration in the LTC used.

We have included a new M&M section (2.1) briefly describing our LTC preparation and citing more detailed methodology therein.

  1. Have the authors checked for any difference in the in vitro bactericidal or virucidal property of PBMC/monocytes after treatment with these LTC?

Although direct assessment of bacteri/virucidal activity is crucial to the validation of LTC immune protection, and we indeed have shown evidence of significant bactericidal activity induced by LTC in canine and bovine macrophages in other studies (references #12 and #22). Subsequent experiments are planned to perform similar experiments in pigs and chickens, but it is currently beyond the scope of the current manuscript.